# The impact of oxytocin on emotion recognition and trust: Does disordered eating moderate these relationships?

Isabel Krug[1]*, Stephanie Fung[1], Shanshan Liu[1], Janet Treasure[2], Chia Huang[3], Kim Felmingham[1], Matthew Fuller-Tyszkiewicz[4,5], Olivia McConchie[1]

**1** Melbourne School of Psychological Sciences, The University of Melbourne, Melbourne, VIC, Australia, **2** Department of Psychological Medicine, King's College London, Institute of Psychiatry and Neuroscience, Section of Eating Disorders, London, United Kingdom, **3** Eating Disorders Program, The Melbourne Clinic, Richmond, Victoria, Australia, **4** School of Psychology, Deakin University, Geelong, VIC, Australia, **5** Centre for Social and Early Emotional Development, Deakin University, Burwood, VIC, Australia

* isabel.krug@unimelb.edu.au

**Data Availability Statement:** The minimal data set for this study is not publicly available due to restrictions posed in the ethics approval. The authors are ethically and legally restricted from

## Abstract

### Objectives

The current study aimed to investigate the impact of oxytocin on emotion recognition, trust, body image, affect, and anxiety and whether eating disorder (ED) symptoms moderated any of these relationships.

### Method

Participants ($n = 149$) were female university students, who were randomly allocated to receive in a double-blind nature, a single dose of oxytocin intranasal spray ($n = 76$) or a placebo (saline) intranasal spray ($n = 73$). Participants were asked to complete an experimental measure of emotion recognition and an investor task aimed to assess trust.

### Results

The oxytocin group exhibited better overall performance on the emotion recognition task (especially with recognising positive emotions), and a decline in state positive affect than the control group at post-intervention. However, these effects were not moderated by ED symptom severity, nor were effects found for state anxiety, negative affect, body image and recognising *negative* emotions in the emotion recognition task.

### Conclusion

The current findings contribute to the growing literature on oxytocin, emotion recognition and positive affect and suggest that ED pathology does not moderate these relationships. Future research would benefit from examining the efficacy of an oxytocin intervention using a within-subjects, cross-over design, in those with sub-clinical and clinical EDs, as well as healthy controls.

sharing the data set because it contains sensitive participant information. The minimal data set that supports the findings of this study may be requested via email (HumanEthics-Enquiries@unimelb.edu.au).

**Funding:** The author(s) received no specific funding for this work.

**Competing interests:** The authors have declared that no competing interests exist.

## Introduction

Individuals with eating disorders (EDs) often have difficulties with aspects of social cognition including emotion recognition [1, 2], theory of mind [3], and trust [4]. It is theorized that difficulties with social cognition [5], and body dissatisfaction [6]–both of which are associated with increased state anxiety and negative mood levels [7]–are involved in the development and maintenance of EDs. Oxytocin has been shown to improve state anxiety and negative mood in healthy individuals [8], and emotion recognition in females with EDs [9]. However, to our knowledge, there have not yet been any studies that have examined the efficacy of an oxytocin intervention on these psychological processes in a female student population, who has been found to be at risk of EDs [10]. Assessing such a population is important since eating pathology occurs on a continuum ranging from severe clinical EDs to subclinical patterns of disordered eating, which would increase the risks of developing full-blown EDs and other mental health disorders if left untreated.

### Emotion recognition, distrust and eating disorders

Several systematic reviews and meta-analyses have demonstrated that individuals with EDs have impaired emotion recognition compared to healthy controls [1, 2, 11]. Furthermore, studies using non-clinical samples have found that individuals with high levels of disordered eating performed worse on emotion recognition tasks [12–14] and reported having more difficulties identifying emotions [15]. Most of these studies used photographs of faces in facial morphing tasks to display different emotions. However, such tasks miss contextual information about real life social interactions. To overcome this limitation, a small number of studies has started to use short video clips to assess emotion recognition in those with EDs (e.g., [16, 17]), and in non-clinical populations with disordered eating (e.g., [13]). Overall, these studies suggest a strong relationship between eating pathology and reduced emotion recognition abilities.

Another prominent and concerning psychological factor contributing to interpersonal difficulties among individuals with eating pathology is interpersonal distrust (i.e., a dispositional tendency to distrust others; [4]), as it has been shown to hinder both social support [18] and the therapeutic alliance [19], thus impeding positive treatment outcomes. Accordingly, ED patients have consistently reported significantly higher levels of interpersonal distrust (as measured by the Eating Disorder Inventory -2 [EDI-2] subscale; [20]) compared to healthy controls (e.g., [21, 22]). Fewer studies have examined the relationship between interpersonal distrust and disordered eating in non-clinical samples of young women [22, 23])—and these studies have reported conflicting results. These divergent findings indicate the need to further explore the link between traits of interpersonal distrust and disordered eating among young women at-risk of developing an ED, such as female university students [10].

### Intranasal oxytocin to improve emotion recognition and trust

One treatment option that has been investigated to address psychiatric difficulties is the administration of oxytocin [24]. Oxytocin is a bonding hormone that has an important role in linking social signals with cognitions and behaviours [25]. Research has indicated that an intranasal oxytocin administration improved emotion recognition and/or state anxiety in healthy individuals [26–29]; and in individuals with social cognitive deficits including autism [30], social anxiety [31], schizophrenia [32] and eating disorders [33, 34].

Only a few published studies have investigated the efficacy of an oxytocin intervention on emotion recognition in individuals with EDs (e.g., [9, 35, 36]). For example, Kim et al. [9] found that, in a sample of female participants, a single dose of intranasal oxytocin (40 IU)

improved overall performance on the Dynamic Facial Morphing Task [37] for healthy controls and patients with BN, especially in detecting sad emotions, but not for individuals with AN. Like Kim et al. [9], the study by Leppanen et al. [35] investigated the impact of 40 IU of oxytocin in 30 females with AN and 29 age-matched healthy women. Results indicated that oxytocin did not significantly influence the interpretation of emotions in either group, which is consistent with Kim et al. [9]'s null findings regarding the AN group.

This null effect in the AN group was also seen in a randomised controlled trial testing intranasal oxytocin in the treatment of AN [36]. However, the non-significant finding regarding the AN group might be due to using an interval (15 minutes) much shorter than the optimal interval of 45 minutes [35, 38] or the use of the 'Reading the Mind in the Eyes Test' (RMET; [39]), which is criticised for having poor validity and reliability [40].

The available literature on the effects of intranasal oxytocin on trusting behaviour is comparatively lacking, and offers mixed results (e.g., [41]). These studies typically used a state or behavioural measure of trust, such as the trust game where participants are randomly assigned to play the role of *investor* or *trustee* and trust is operationalised as the investment amount (i.e., higher investment corresponds to greater trust) [42–45].

Kosfeld et al. [44] and Baumgartner et al. [42] found that, among healthy males, the average trust game investment amount was greater in an oxytocin group than in a placebo group. More recently, Declerck et al. [46] attempted to replicate these findings but failed to produce the same results. However, the exploratory analyses revealed that oxytocin may increase trust in individuals with a lower depositional level of trust in a no-social-contact condition [46]. As most existing studies assessing the oxytocin-trust relationship have been conducted in non-clinical male samples, it remains unclear how this relationship relates to a young female sample. Further empirical research is therefore warranted to elucidate the oxytocin-trust relationship in non-clinical female samples who might be at risk for EDs.

## Gaps in the literature

Although there has been growing enthusiasm for the therapeutic potential of intranasal oxytocin for addressing a variety of social-emotional difficulties in EDs (e.g., [9]), research has been scarce and the results inconsistent (e.g., [9, 35, 36, 44, 46]). Such conflicting findings could be due to the different emotion recognition tasks used. Video clips, such as the Emotion Evaluation Test of the Awareness of Social Inference Test–Revised (TASIT-R; [47]) have been found to provide more contextual information of emotions than morphing [37] or the 'Reading the mind in the eyes' [39] tasks but have not yet been used when assessing the effects of oxytocin on emotion recognitions. Furthermore, research examining the effects of intranasally administered oxytocin on trusting behaviour has also produced mixed results [44, 46]. To our knowledge no study has yet examined the potential trust- and affect-altering effects of intranasal oxytocin in the context of eating pathology. These gaps in the literature are important to address since assessing and addressing social emotional difficulties (e.g., emotion recognition and trust) in relation to eating pathology could inform early prevention and intervention strategies for EDs.

## The current study

The current study aimed to extend previous research and fill in current gaps in the literature by comparing two conditions (placebo and oxytocin) and performance on an emotion recognition task using video clips (TASIT-R; [47]) and a trust game in a female university sample, considered to be at-risk for EDs [10]. State anxiety, positive and negative affect and body satisfaction were assessed before and after individuals received either the placebo or oxytocin

intervention, as these have been shown to be related to disordered eating [7] and were not previously assessed by Kim et al. [9] and Leppanen et al. [35]. Specifically, given the co-occurrence of increased body dissatisfaction and state anxiety [7] in females with high levels of disordered eating and that oxytocin has been shown to decrease state anxiety [8], the current study theorised that if state anxiety decreased following an oxytocin intervention state body satisfaction would increase.

We therefore hypothesised that compared to the placebo condition, (Hypothesis 1) participants in the oxytocin condition would score higher on emotion recognition and increased trust, and (Hypothesis 2) participants in the oxytocin condition would report decreased state anxiety and negative affect and increased state body satisfaction at post-intervention assessment. We also assumed that the effects in 1 and 2 would be most pronounced for the at-risk group for EDs compared to a non-at-risk ED group (Hypothesis 3).

## Method

### Design overview

This study used a randomised double-blind, placebo-controlled, between-groups research design. Participants attended a single laboratory session for approximately 2.5 hours, in which they received either an intranasal oxytocin spray (oxytocin group) or intranasal saline spray (placebo group) and completed all self-report and experimental measures (see Materials and Procedure).

### Participants

Participants were 149 female undergraduate psychology students recruited through a Research Experience Program (REP) at a university in Melbourne from Oct 2016 to Dec 2017, who took part in exchange for 2.5 hours' worth of course credit. Seventy-three participants were randomly allocated to the oxytocin group and 73 to the placebo group.

Exclusion criteria (***self-reported***) included: a.) males; b.) aged <18 or >30; c.) being pregnant/trying to become pregnant; d.) breast feeding/breast-fed in the last two months; e.) heavy smokers (> 5 cigarettes per day) and/or heavy drinkers (>14 standard drinks per week); f.) having a current substance use disorder or any other psychiatric condition other than an ED and g.) a history of heart disease, cardiac disorder, neurological disorder, or allergic reaction to preservatives (food additives). Exclusionary criteria were selected based on safety reasons (i.e., oxytocin can induce contractions in pregnancy) and on findings that these factors can interfere with the effectiveness of oxytocin [48].

Participants were also asked to disclose their use of current medications (including oral contraceptives), herbal products and supplements; they were excluded if the medical advisor (a psychiatrist working at an ED Unit at the Melbourne Clinic) indicated safety concerns (i.e., an interaction between their medication and oxytocin).

Participants were asked to abstain from caffeine and alcohol in the 24 hours prior to the laboratory visit, and refrain from drinking (water was allowed) or eating food in the 2 hours prior to the laboratory visit, as these can influence the effect of oxytocin [9]. This study was approved by Behavioural and Social Sciences Human Ethics Sub-Committees (HESC) at the University of Melbourne (ethics application no.1647944).Written informed consent was obtained from each participant.

### Materials

**Screening measure.** A screening questionnaire, based on exclusionary criteria, was used to assess the suitability of potential participants.

**Pre- and post-intervention state measures.** *State-Trait Anxiety Inventory, Form Y-1 (STAI-Y, [49]).* The STAI-Y is a reliable and valid measure of self-reported state anxiety, commonly used in clinical and research settings [50]. The STAI-Y consists of 20 items, assessing state anxiety, as opposed to long-standing personality trait anxiety. Participants rate items such as "I am worried" on a 4-point scale from 1 = "not at all", to 4 = "very much so". Total scores can range from 20–80, with higher scores indicating greater levels of state anxiety. In the current sample, the Cronbach's alphas at the two time points were 0.69 (T1) and 0.77 (T2) for the placebo group, and 0.68 (T1) and 0.59 (T2) for the oxytocin group.

*Positive and Negative Affect Schedule (PANAS; [51]).* The PANAS consists of two 10-item mood scales assessing states of positive affect (e.g., "Excited") and negative affect (e.g., "Distressed"). Participants indicated the degree to which they felt each emotion at the present moment on a 5-point rating scale (1 = Very slightly or not at all; 5 = Extremely). Responses were summed to derive separate total scores for positive affect and negative affect; higher scores indicate higher levels of each respective affective state. In the present study, the Cronbach's alphas were 0.86 (T1) and 0.90 (T2) for positive affect, and 0.79 (T1) and 0.90 (T2) for negative affect for the placebo group; for the oxytocin group, the Cronbach's alphas were 0.90 (T1) and 0.93 (T2) for positive affect, and 0.88 (T1) and 0.79 (T2) for negative affect.

*Body Image States Scale (BISS; [52]).* The BISS is a 6-item scale that assesses an individual's evaluation of their physical appearance in the present moment. Participants select the statement that best describes how they feel on a 9-point rating scale (e.g., "Right now I feel . . . 'Extremely dissatisfied with my body size and shape' = 1, 'Extremely satisfied with my body size and shape' = 9). Total scores can range from 6–54, with lower scores indicating more negative body image states (body dissatisfaction), and higher scores indicating more positive body image states (body satisfaction). The Cronbach's alphas were 0.80 (T1) and 0.77 (T2) for the placebo group, and 0.75 (T1) and 0.80 (T2) for the oxytocin group.

**Intranasal spray (intervention).** The intranasal sprays, which were both odorless, were contained in identical bottles with labels stating the participant code (i.e., 1, 2, 3 etc.). This number was linked to the condition (i.e., oxytocin or placebo intervention). Participants received 3 puffs in each nostril, 6 puffs in total. The placebo spray (control condition) was normal saline. The oxytocin spray (24 IU of Oxytocin, Syntocinon®, manufactured by Novartis Pharmaceutical Australia Pty Ltd) was considered a standard, safe, and optimal dose for a single intranasal oxytocin administration [38, 48]. Note that the participants and researchers were blind to the condition type. The project supervisor and principal researcher (IK), who did not collect and/or analyse the data, randomly assigned participants to their condition (oxytocin or placebo) using an online number generator.

**Immediately Post-Intervention Questionnaires.** *Demographic and Clinical History Questionnaire.* Demographic information including age, ethnicity, main language spoken at home, education level, employment status, marital status, and self-reported height and weight (used to calculate body mass index, BMI) were collected. Clinical information including current medication use (including contraceptive medication), timing of menstrual cycle, past psychiatric history, and history of ED were also collected.

*Eating Attitudes Test (EAT-26, [53]).* The EAT-26 is a 26-item scale that assesses eating disorder symptoms. Participants rate items such as "I am terrified about being overweight" on a 6-point scale to indicate the frequency with which the statement applies to them (3 = Always, 2 = Usually, 1 = Often, 0 = Sometimes, 0 = Rarely, 0 = Never). Total scores can range from 0–78, with scores 20 and above indicating a high level of concern regarding problematic eating behaviour, body weight, or dieting. The EAT-26 has been found to be a valid and reliable measure [53], with excellent internal consistency in the current sample. In the current study the Cronbach's alphas were 0.88 for the placebo and 0.86 for the oxytocin group.

### Time-filler task

Participants watched a national geographic documentary on sea-life for approximately 25 minutes on their own computer screen with headphones. The video was the same for all participants and selected as a neutral activity.

**45 minutes post-intervention experimental measures.** *Emotion Recognition Task (TASIT-R; [47]).* The TASIT-R task involves participants watching 28 short video clips (15–60 seconds) of social interactions displaying the 6 primary emotions (anger, disgust, fear, happiness, sadness, and surprise) and neutral affect. Participants wear headphones and are presented the video clips on their own computer screen; the accuracy of selecting the correct emotion is assessed. The total emotions correct, the total positive emotions correct (happy, surprised, neutral) and the total negative emotions correct (sad, angry, anxious, disgust) were calculated for each participant. The following range of correct scores are possible: total emotions: 0–28; positive emotions: 0–12, and negative emotions: 0–16. Higher scores indicate better emotion recognition skills. The TASIT-R has been shown to have acceptable reliability and validity [47], and excellent internal consistency in the current sample.

*Experimental Measure of Trusting Behaviour (The trust game; [43]).* The trust game was used as an experimental measure of trusting behaviour at T2. At the beginning of the game, each participant read a passage of text that was designed to: (a) provide instructions for how to play as the investor player; and, (b) deceive participants to believe that they were playing with a counterpart trustee player, when in fact their responses were not linked to any other participant (as per previous research; [54]). Participants only played the investor (i.e., greater investments reflect greater trusting behaviour).

The instructions indicated that both players had received an initial endowment of 50 experimental monetary units (EMU)—no real payment was included in this study. Participants were informed that as an investor, they had the opportunity to send any amount of their EMU endowment (i.e., 0–50) to their counterpart, and that this amount would be automatically tripled during the transaction process. Following the transfer, the trustee then had the opportunity to return some, all, or none of the total EMU in their possession back to the investor.

**Post experiment survey.** A brief survey asked participants to list any side effects or allergic reactions they were experiencing.

### Procedures

Testing occurred during the daytime in a quiet computer laboratory; participants were seated in front of their own computer, with two student researchers present for the duration of the 2.5-hour study. The questionnaires were run using the online program, Qualtrics. Participants firstly read a Plain Language Statement and then completed the screening questionnaire. Those deemed eligible signed a consent form; they were then assigned a code (i.e., 1, 2, 3 etc.), linked to the condition (oxytocin or placebo) they would receive. The participants and the investigators (O.M and S.F) involved in data collection were blinded. Only, the responsible researcher (I.K) was un-blinded in case a participant reported an allergic reaction or side effect.

Participants firstly completed the pre-intervention (baseline) state measures (STAI-Y, PANAS and BISS), with the order counterbalanced to control for order effects. Participants were then instructed on how to use the intranasal spray and self-administered one dosage (24 IU) of either the oxytocin or placebo nasal spray, in front of the researchers. Next, participants completed the immediately post-intervention measures, involving the demographic questionnaire and the trait measures EAT-26 (considered independent from the effects of the intervention). Following this, participants watched a 25-minute documentary on sea-life, which

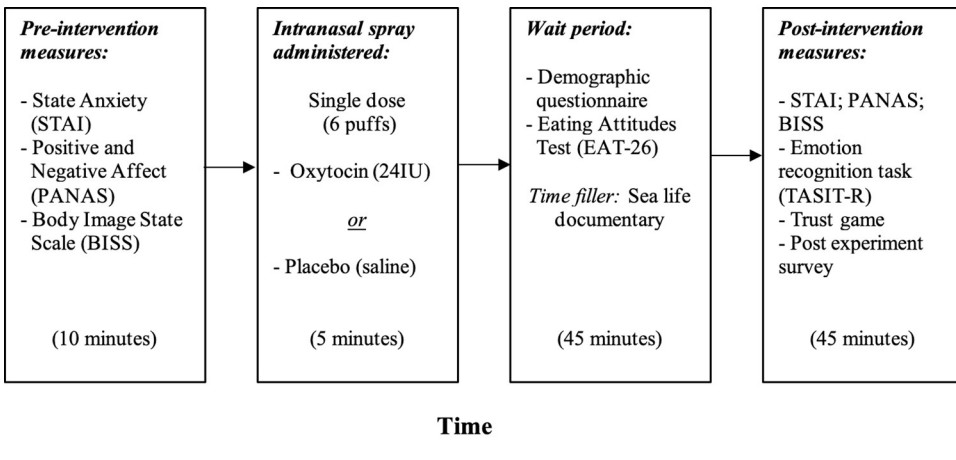

**Fig 1. Procedure for the laboratory visit.**

resulted in a 45-minute interval for the intranasal spray to take effect (this interval was selected based on findings by Spengler et al., [38]). After participants watched the sea-life documentary, they then completed the post-intervention state measures (STAI-Y, PANAS and BISS; order again counterbalanced) and the experimental measure of emotion recognition (TASIT-R) and the investor game. The TASIT-R and trust investor game were administered exclusively post-intervention, rather than at baseline, to prevent potential effects on these tasks from confounding the actual effects of the oxytocin or saline nasal sprays. Lastly, participants completed the post experiment survey. The procedure for the laboratory visit is outlined below in Fig 1.

At the conclusion of the study, participants were provided with a debrief form, which outlined the aims of the study, and contained information regarding monitoring for possible side effects, and a list of support services. Participants were also provided with an envelope, which notified them which intervention (oxytocin or placebo) they were allocated to. Note that the student researchers (O.M and S.F), were not un-blinded until after data analysis. A seasoned statistician (M.F), who was un-blinded, then verified these analyses for publication purposes.

Participant total scores on the EAT-26 were calculated on the day of the study. Those scoring 20 or above, suggesting a high level of concern regarding bodyweight, dieting, or eating behaviours [53], were contacted via email within 24 hours by the project supervisor (IK), who provided them with information on a range of appropriate support services.

## Data analysis

The data were analysed using SPSS Statistics V.23. Initial analyses were first conducted in 2017/2018 for two independent Master theses (O.M and S.F.) and later re-run for the current article in 2022 by a seasoned statistician (M.F). An independent samples t-test was used to test the first hypothesis (i.e., that compared to the placebo condition, participants in the oxytocin condition would perform better on an emotion recognition task and a trust game), by comparing the two groups (oxytocin and placebo) on the emotion recognition task (TASIT-R) and their investment amounts in the trust game.

The second hypothesis (i.e. that compared to the placebo condition, participants in the oxytocin condition would report decreased state anxiety and increased state body satisfaction at post assessment) was analysed using a 2 (Group: oxytocin and placebo) x 2 (Time point: pre and post intervention) mixed between-within subjects analyses of variance (ANOVA), with

state anxiety (STAI-Y scores) and state body satisfaction (BISS scores) as the outcome variables in cases where there was a baseline assessment, scores at T1 were used as covariates.

The third hypothesis (i.e., that ED symptoms would moderate the relationship between group [oxytocin or placebo] and emotion recognition performance) was assessed using a simple regression-based moderation analysis (model 1) to retain ED symptom severity as a continuous variable. Cohen's d was used as an effect size measure, with d = 0.2 considered as "small", 0.5 as "medium" and 0.8 as "large" effect sizes [55].

Sample size targets for the present study were based on assumption of a moderate between-group difference for experimental and control conditions (consistent with calculations of [9] for our outcome measures. Setting alpha at .05 (two-tailed), power = .80, and Cohen's d > .45, we set a target of 75 per arm (150 total sample).

Although a range of outcome measures was tested, we retained alpha criterion at .05 rather than correcting for multiple comparisons. Given the paucity of studies exploring the effects of oxytocin in this context, there is a risk that correction for Type I error rate increases risk of Type II errors [56–58], prematurely terminating a viable line of inquiry.

## Results

### Frequency statistics

Table 1 shows the frequency statistics and test statistics for the demographic and clinical variables, additionally to high EAT-26 total scores (i.e., scores ≥ 20) for the overall sample, and the oxytocin and placebo groups. Means and standard deviations for age and BMI are also presented in Table 1.

Table 1 shows that the two groups were generally equivalent at baseline; chi-square tests confirmed no significant differences between the oxytocin and placebo groups, for the demographic (ethnicity, main language spoken at home, paid employment, education, marital status, and clinical variables (use of the contraceptive pill, other medication, self-reported psychiatric diagnoses, current and lifetime ED, day 3 to12 of menstrual cycle). However, participants in the oxytocin group scored significantly higher on eating pathology than the control group (32% vs 15%). Independent samples t-tests revealed no significant differences between groups for age and BMI.

Fourteen percent of the sample (N = 21) had a high EAT-26 total score, and 10% of the sample (N = 15) reported having a history of an ED, with similar ratios across the two groups. There were only 3 participants with a lifetime self-reported psychiatric condition other than an ED (2 with mixed depression/anxiety and 1 with anxiety). Only 22% of the overall sample were currently using the contraceptive pill and 29% were currently on day 3 to 12 of their menstrual cycle.

Table 2 presents the means, standard deviations, and test statistics across the groups.

### Differences in outcome variables between the oxytocin and control group at pre-intervention

Independent-samples t-tests indicated no significant differences between groups in terms of pre-intervention BISS, PANAS and STAI-Y scores, and on the trait measure (EAT-26 score), suggesting the groups were equivalent in terms of state anxiety, state body satisfaction, ED symptoms, prior to nasal spray administration (see Table 2 for means and standard deviations) (all *p*s ranged from .447 to .951, Cohen's d ranged from .01 to .13).

### Hypotheses testing

**Hypothesis 1—Oxytocin and Better Performance on the Recognition of Positive Emotions.**   There was a significant intervention group effect for the TASIT-R total score (*t*

**Table 1. Demographic and clinical variables of the sample, oxytocin and placebo groups.**

| Demographics Categorical Variables | | Total sample | | Oxytocin group | | Placebo group | | Test statistics | | |
|---|---|---|---|---|---|---|---|---|---|---|
| | | (N = 149) | | (n = 76) | | (n = 73) | | | | |
| | | n | % | n | % | n | % | χ² | df | P |
| Ethnicity | White/Caucasian | 53 | 36 | 27 | 36 | 26 | 36 | 0.13 | 2 | .94 |
| | Asian/Mid. Eastern | 69 | 46 | 36 | 47 | 33 | 45 | | | |
| | Other[1] | 27 | 18 | 13 | 17 | 14 | 19 | | | |
| Main language spoken at home | English | 100 | 67 | 47 | 62 | 53 | 73 | 1.95 | 1 | .16 |
| | Other[2] | 49 | 33 | 29 | 38 | 20 | 27 | | | |
| Paid Employment | Part time/casual | 64 | 43 | 36 | 47 | 28 | 38 | 1.23 | 1 | .27 |
| | Unemployed | 85 | 57 | 40 | 53 | 45 | 62 | | | |
| Education (highest level) | Year 12 or equiv. | 124 | 83 | 62 | 82 | 62 | 85 | 0.31 | 2 | .86 |
| | University qualification[3] | 18 | 12 | 10 | 13 | 8 | 11 | | | |
| | Other[4] | 7 | 5 | 4 | 5 | 3 | 4 | | | |
| Marital status | Never married | 149 | 100 | 76 | 100 | 73 | 100 | - | - | - |
| | Married | 0 | 0 | 0 | 0 | 0 | 0 | | | |
| Using oral contraceptive medication | Yes | 32 | 22 | 14 | 18 | 18 | 25 | 0.86 | 1 | .35 |
| | No | 117 | 78 | 62 | 82 | 55 | 75 | | | |
| Using medication[5] | Yes | 8 | 5 | 4 | 5 | 4 | 6 | 0.00 | 1 | .95 |
| | No | 141 | 95 | 72 | 95 | 69 | 94 | | | |
| History of eating disorder[6] | Yes | 14 | 9 | 7 | 9 | 7 | 10 | 0.01 | 1 | .94 |
| | No | 135 | 91 | 69 | 91 | 66 | 90 | | | |
| Current eating disorder | Yes | 1 | 1 | 1 | 1 | 0 | 0 | 0.97 | 1 | .33 |
| | No | 148 | 99 | 75 | 99 | 73 | 100 | | | |
| History of psychiatric condition | Yes | 3 | 2 | 2 | 3 | 1 | 1 | 0.30 | 1 | .58 |
| | No | 146 | 98 | 74 | 97 | 72 | 99 | | | |
| On day 3–12 of menstrual cycle | Yes | 43 | 29 | 24 | 32 | 19 | 26 | 1.16 | 2 | .56 |
| | No | 53 | 36 | 28 | 37 | 25 | 34 | | | |
| | Don't know or N/A | 53 | 36 | 24 | 32 | 29 | 40 | | | |
| High EAT-26 total score (≥ 20) | Yes | 35 | 24 | 24 | 32 | 11 | 15 | 5.65 | 1 | .02 |
| | No | 114 | 76 | 52 | 68 | 62 | 85 | | | |
| Continuous Variables | | M | SD | M | SD | M | SD | t (df) | p | d |
| | Age | 18.79 | 1.08 | 18.9 | 1.08 | 18.7 | 1.08 | .96 (147) | .34 | .16 |
| | BMI | 17.75 | 2.89 | 17.9 | 3.15 | 17.6 | 2.60 | .57 (147) | .57 | .09 |

*Note.* df = degrees of freedom; *d* = Cohen's d.

[1] "Other" combined languages including Chinese, Mandarin, Cantonese, Japanese, Vietnamese, Tamil, Indonesian, Formosan, and Norwegian.

[2] "Other" combined a minority of participants who identified as Aboriginal or Eurasian.

[3] "University Qualification" included Bachelor degree and Advanced Diploma.

[4] "Other" included certificate and still in high school.

[5] "Medication" included panadol, and herbal products and supplements such as Vitamin C, B12 and D, iron tablets, multivitamins, echinacea, and probiotics.

[6] "History of eating disorder" included Anorexia Nervosa, Bulimia Nervosa, Binge Eating Disorder, and EDNOS

($df$ = 147) = 2.02, $p$ = .02, Cohen's $d$ = .33) and the TASIT-R positive subscale ($t$ ($df$ = 147) = 2.39, $p$ = .01, Cohen's $d$ = .39). As shown in Table 2, post-intervention scores were higher for the oxytocin group than for the placebo condition. Neither the TASIT-R negative subscale score ($t$ ($df$ = 147) = 1.18, $p$ = .12, Cohen's $d$ = .19) nor the trust investor game ($t$ ($df$ = 147) = 0.66, $p$ = .26, Cohen's $d$ = .11) exhibited intervention group effects.

**Hypothesis 2 –Oxytocin and Decreased State Anxiety (STAI-Y) and Affect (PANAS) and Body Satisfaction (BISS).** There was a significant time x group effect for PANAS positive affect scores ($F$ (1, 147) = 4.04, $p$ = .046, $n_p^2$ = .03), but not for PANAS negative affect scores ($F$ (1, 147) = 0.29, $p$ = .5, $n_p^2$ < .01). As shown in Table 2, positive affect scores decreased for both groups post- intervention, but more so for the oxytocin group. Finally, there was no

**Table 2. Means, standard deviations, and test statistics for all measures.**

| Measure | Time | Oxytocin group (n = 76) | | Placebo group (n = 73) | |
|---|---|---|---|---|---|
| | | M | SD | M | SD |
| Body Satisfaction: | 1 (pre) | 30.84 | 7.06 | 29.93 | 7.53 |
| | 2 (post) | 31.67 | 7.29 | 31.45 | 7.05 |
| State Anxiety: | 1 (pre) | 38.14 | 8.56 | 37.78 | 8.87 |
| | 2 (post) | 38.25 | 8.86 | 38.77 | 9.17 |
| PANAS positive | 1 (pre) | 25.26 | 7.65 | 25.19 | 6.54 |
| | 2 (post) | 20.08 | 8.03 | 22.10 | 7.21 |
| PANAS negative | 1 (pre) | 13.89 | 4.76 | 13.64 | 3.79 |
| | 2 (post) | 12.04 | 2.91 | 12.11 | 3.92 |
| Eating Pathology: | - | 10.05 | 9.47 | 10.79 | 10.67 |
| Trust Investor | (post) | 30.28 | 12.17 | 28.89 | 13.55 |
| Emotion Recognition: TASIT | | | | | |
| *Total emotions correct* | | 25.30 | 1.96 | 24.56 | 2.49 |
| *Positive emotions correct* | | 10.80 | 0.91 | 10.40 | 1.15 |
| *Negative emotions correct* | | 14.50 | 1.52 | 14.16 | 1.94 |

*Note.* N = 149; df = degrees of freedom; d = Cohen's d; p < 0.5 (two-tailed); Time 1 (pre-intervention), Time 2 (post-intervention). TASIT positive emotions included happy, surprised, and neutral; TASIT negative emotions included sad, angry, disgusted, anxious.

time x group interaction effect for state body satisfaction ($F(1, 147) = 1.05$, $p = .31$, $n_p^2 = .01$) or state anxiety scores ($F(1, 147) = 0.47$, $p = .49$, $n_p^2 < .01$).

**Hypothesis 3 –Moderation effects of eating pathology.** Eating pathology (EAT-26 scores) did not moderate any group effects mentioned above ($ps > .05$), with the exception of state anxiety as an outcome (see Table 3). Though the experimental oxytocin group had a lower state anxiety score post- intervention relative to the control group, this effect was weaker for individuals with elevated eating pathology.

## Discussion

The current study compared participants in two conditions (oxytocin and placebo) and their performance on an emotion recognition task (TASIT-R; video clips) and a trust investor game while also assessing the moderating effects of ED symptoms on the oxytocin-emotion

**Table 3. Moderating effects of eating disorder pathology on main effects.**

| Outcome | F* | p | ΔR2 |
|---|---|---|---|
| Body satisfaction | 1.23 | .270 | .003 |
| State anxiety | 8.40 | .004 | .034 |
| PANAS positive | 0.01 | .941 | .000 |
| PANAS negative | 1.53 | .218 | .007 |
| Trust investor | 1.19 | .277 | .008 |
| TASIT total | 0.26 | .611 | .002 |
| TASIT positive | 0.28 | .595 | .002 |
| TASIT negative | 0.04 | .835 | .000 |

Notes

*df = 1, 145 for change with introduction of interaction term in regression model.

recognition/trust relationship. The study also compared participant scores on state anxiety, affect and state body satisfaction measures before and after participants received an intranasal spray of either oxytocin or placebo. Participants in the two groups were equivalent at baseline in terms of age, education level, BMI, state anxiety, affect and state body satisfaction.

Our first hypothesis was only partially supported. Compared to the placebo condition, participants in the oxytocin condition exhibited overall a better performance on the emotion recognition task (TASIT-R) and scored higher on the recognition of positive emotions. However, no significant differences between the intervention groups were found for correctly identifying negative emotions. Furthermore, there was no significant difference between groups regarding performance on the investor trust game. Similarly, the second hypothesis that compared to the placebo condition, participants in the oxytocin condition would report decreased state anxiety, negative affect, and increased state body satisfaction at post-intervention, was not supported. However, contrary to expectations, positive affect declined post- intervention, and this decline was more pronounced in the oxytocin than the control group. Finally, ED symptoms did not moderate any of the relationships discussed above.

## Emotion recognition

To our knowledge this was the first study to examine differences in emotion recognition using the TASIT-R after having been exposed to an oxytocin/placebo intervention in the context of (non-clinical) disordered eating. Our findings suggest that the oxytocin administration was associated with better performance on the TASIT-R, especially in identifying positive emotions. This finding is broadly in line with Kim et al.'s [9] findings, who also found that compared to a placebo condition, a single dose of intranasal oxytocin produced a significant increase in overall emotion recognition skills in females with BN and in healthy female controls (although the effect was not found in the AN group). Our finding is also consistent with research involving healthy individuals [28], which reported that a single dose of oxytocin administration improved emotion recognition performance in both male and female participants [9, 29].

Together, these findings suggest that the effect of oxytocin on emotion recognition performance may be extended beyond clinical samples to a student population who has been found to be at risk for an ED [10]. It is important to note, however, that our finding regarding the recognition of negative emotions (null effects for the oxytocin group) contradicts previous research reporting increased sensitivity for detecting negative emotions (such as sadness and fear) in participants with BN or healthy controls after receiving an intranasal oxytocin intervention [9, 59].

The overall contradicting finding regarding the type of emotion (positive versus negative) is in line with the more general emotion recognition research in EDs, which has outlined that there is currently no consensus on emotion recognition deficits in relation to specific emotions in the context of disordered eating [12, 13]. This may indicate that recognising specific types of emotions is complex and individualised, and further investigation in this area is therefore needed.

## Trusting behaviour

Contrary to prediction, there was no significant difference between displays of trusting behaviour in the oxytocin and placebo groups, indicating that there was no general effect of intranasal oxytocin on trusting behaviour in this nonclinical sample of young women. This result contradicts earlier studies that reported increased trusting behaviour after intranasal oxytocin administration in nonclinical samples [42, 44]. However, the null finding is in line with later

studies [46] who failed to replicate findings of Kosfeld et al. [44]. Taken together, the more recent findings seem to show that intranasal oxytocin does not necessarily improve trusting behaviour.

One possible explanation for the null effects may be related to the context of the oxytocin administration. Existing research indicates that the context in which intranasal oxytocin is administered can have differential results [60, 61]. Specifically, it was suggested that oxytocin may increase perceptual salience of social information, which in turn, influences behavioural responses (e.g., [61]). In other words, social information is crucial to "trigger" oxytocin's effects. Different from the present study, participants in past research indeed had access to a considerable amount of social information. For instance, Kosfeld et al. [44] allowed participants to interact during the wait-period following the oxytocin administration, and Baumgartner et al. [42], and Yao et al. [62], provided participants with varying types of feedback about their anonymous counterpart's back-transfer history. In contrast, participants in the present study were provided with limited social information (e.g., no social interaction or trust game feedback) to minimise presumed confounding effects (i.e., attribution of trusting behaviour to influences other than oxytocin). Against the background of the reviewed research, however, social information might well be necessary to initiate oxytocin's trust-promoting effects, with its relative absence possibly contributing to present null findings.

## State anxiety, affect and state body satisfaction

Unexpectedly, there was no effect of the intranasal oxytocin intervention on either state anxiety, state negative affect or body satisfaction. The null finding on state anxiety opposed previous experimental research that reported decreased state anxiety as measured on the STAI-Y in healthy males after an intranasal oxytocin administration [8, 63]. Differences in methodology may explain the null findings in the current study. For instance, Goodin et al. [8] used a within-subjects cross-over design, meaning participants were used as their own controls; whereas in the current study, a between-subjects design was used; meaning between group differences could not be controlled for (note this is discussed further in the limitations section). While results from the current study vary from past research investigating oxytocin and state anxiety in healthy adults, scarce research has investigated oxytocin and state anxiety in the context of eating pathology, thus it is difficult to interpret the null finding for state anxiety further.

Interestingly, contrary to prediction, intranasal oxytocin decreased state positive affect relative to a placebo. To our knowledge, this study was the first to investigate the effects of intranasal oxytocin on positive affect, and its results contradict the supposition that oxytocin produces positive affective states concomitant to decreases in negative affective states. Although surprising, intranasal oxytocin has been shown to increase meta-mood (awareness of one's emotions) in young women [64], suggesting one possible mechanism underlying the current finding is that participants in the oxytocin group were more aware of their diminished positive affect, and thus reported greater decreases compared to participants in the placebo group. Nonetheless, if this was the case, it is difficult to interpret why participants in the oxytocin group did not also report significantly greater decreases in negative affect. In view of the probable influence of context on oxytocin's effects (e.g., [60]), an alternative mechanism might be that intranasal oxytocin sensitised participants to available environmental cues in the experiment [65]. Specifically, oxytocin might have increased the salience of undesirable features of the situation, such as the lack of social interaction over a considerable period (2.5 hours), in turn, dampening positive affect. Nevertheless, future research is needed to verify the mechanisms proposed here and/or clarify other mechanisms underpinning the impeding effect of intranasal oxytocin on state positive affect found in the present study.

## No moderating effects of eating pathology

The finding that ED symptoms did not have a moderating effect on the oxytocin-emotion recognition relationship was contrary to predictions based on previous research. Given that previous findings in non-clinical samples suggested that those with higher levels of disordered eating had greater deficits in emotion recognition [12–14], the current study theorized that they could benefit most from oxytocin. However, the results in the current study suggested that the level of ED symptoms did not impact emotion recognition performance after the oxytocin intervention. It is important to note though that these previous studies [12–14] did not involve an oxytocin intervention; they only examined emotion recognition in the context of disordered eating; thus, it is difficult to compare the studies in terms of the moderating effect of disordered eating on the oxytocin-emotion recognition relationship.

The level of eating pathology did also not moderate the relationship between the oxytocin intervention and performance on the investor trust game or the other state variables (i.e., state anxiety, positive and negative affect; body satisfaction). These null findings might indicate that there are no amplified trust-altering effects of intranasal oxytocin for young women for whom traits of disordered eating are below clinical thresholds. Although oxytocinergic dysfunction has been observed in EDs [66, 67], it is unclear whether this is a cause or consequence of eating pathology and ongoing interpersonal difficulties. It is possible that oxytocinergic dysfunction is only relevant to clinical EDs, and hence intranasal oxytocin will not confer corrective, amplified effects on trusting behaviour in healthy young women with high, but not clinically indicated, traits of disordered eating or interpersonal distrust.

## Limitations and future directions

The current study should be interpreted within the context of several limitations. First, the study did not incorporate a within-subjects cross-over design, as per previous research investigating oxytocin, emotion recognition and EDs [9, 35]. Individual differences between groups were therefore not controlled for, which may have contributed to the null findings. The primary reason for having a between-subjects design in this study was due to time constraints (REP participants cannot be involved in studies >2.5 hours, hence it was not possible to have them complete all measures on two separate occasions). Future research could overcome this limitation by using a sample not restricted by time constraints, thus allowing the design to be a within-subjects, cross-over design.

The second main limitation of the study was that the emotion recognition and trust tasks were not completed before and after the oxytocin administration. Hence, it was not possible to examine pre- and post-intervention performance on these tasks, which could have contributed to the null findings. This decision was because practice effects on these tasks could have interfered with the intervention effects and due to time constraints. Future research could address this limitation by incorporating a repeated measures design for the TASIT-R and the trust investor game. Please note that the TASIT-R is available as a repeated measures design, with variations in the video clips shown to control for practice effects [47]. However, such variations currently do not exist for the trust investor game.

Third, it was likely that the emotion recognition and trust tasks were not sensitive enough for the sample utilised (as previously discussed). While research suggests that those with disordered eating have impaired emotion recognition (i.e., [12–14]) and trust (i.e., [22, 23]) skills, those in the current study who scored highly on the EAT-26 (suggestive of disordered eating; [53]) were also first year psychology students, and therefore likely to have an interest in emotions and interpersonal relationships, and perhaps possessed good emotion recognition and interpersonal skills. This may have created ceiling effects in the current study. Future research

could overcome this limitation by utilising a young female community sample, or female students from outside of the psychology school. Alternatively, future research could incorporate a timed component in the TASIT-R and investor game (i.e., measure response time and accuracy of the task performance), as this could increase its sensitivity for detecting skill differences between participants.

Fourth, it is important to acknowledge that the perception of emotions, such as surprise, can exhibit considerable variability contingent upon the circumstances. For instance, surprise may evoke positive sentiments when linked with delightful news, whereas it may evoke negative reactions when associated with shock or fear. Similarly, neutrality can be construed positively, reflecting a sense of calmness, or negatively, suggesting indifference. Despite the comprehensive validation of the TASIT-R for assessing social perception, it may not fully capture these intricate nuances, primarily due to the subjective and context-dependent nature of emotional responses.

Fifth, while the observed effect sizes and statistically significant results for the emotion recognition and state positive emotion findings are promising, it is important to approach these findings with caution. The possibility of Type I errors, arising from multiple comparisons, necessitates replication of these results in further studies to confirm their validity.

Sixth, we used a non-clinical sample, thus reducing the generalisability of findings to clinical populations. Future research could therefore incorporate another group of female participants with diagnosed EDs (AN and BN). This would allow for comparisons between those with disordered eating (sub-clinical), those with diagnosed EDs, and healthy controls, in the context of oxytocin and emotion recognition.

Seventh, the choice of saline as a placebo in this study may introduce a further limitation, as it may not completely replicate the inert properties of a placebo containing all inactive ingredients. Future studies should therefore employ placebos containing all inactive ingredients to mitigate potential subjective effects of substances other than oxytocin.

Finally, our sample were young females of whom 22% took oral contraceptives. It should be noted that women who take oral contraceptives may sustain elevated oxytocin levels over prolonged periods. This heightened presence of oxytocin could have diminished the impact of the moderate releases of oxytocin in our oxytocin group, potentially altering its usual physiological effects in this group [68]. Similarly, it should be acknowledged that oxytocin may vary throughout the menstrual cycle, in tandem with shifts in the concentrations of female sex hormones and that these fluctuations could have impacted our findings [69].

## Conclusion

This was the first study to investigate the impact of oxytocin on emotion recognition, trusting behaviour, affect, anxiety and body satisfaction, and the moderating effects of ED symptoms, by comparing participants randomly allocated to receive a single dose of either 24 IU of oxytocin or placebo intranasal spray. The findings suggest that participants in the oxytocin condition perform better on the emotion recognition task compared to the control group, especially in terms of recognising positive emotions. However, these effects were not found for recognising negative emotions nor for trusting behaviour. Furthermore, only state positive affect was impacted by the oxytocin intervention, but not state anxiety, negative affect, or body image states. ED symptom severity did not moderate the relationships proposed above. Future research could examine the efficacy of an oxytocin intervention using a within-subjects, crossover design, in those with sub-clinical and clinical EDs, and healthy controls to better understand its therapeutic potential for enhancing social-emotional functioning in the context of eating pathology.

## Acknowledgments

We thank Viola Puccio, Hayley Brown, and Mercedes Delgado Arroyo for their valuable contributions for the data collection.

## Author Contributions

**Conceptualization:** Isabel Krug, Stephanie Fung, Kim Felmingham, Olivia McConchie.

**Data curation:** Isabel Krug, Stephanie Fung, Chia Huang, Olivia McConchie.

**Formal analysis:** Matthew Fuller-Tyszkiewicz.

**Investigation:** Isabel Krug, Stephanie Fung, Chia Huang, Olivia McConchie.

**Methodology:** Isabel Krug, Stephanie Fung, Kim Felmingham, Olivia McConchie.

**Software:** Matthew Fuller-Tyszkiewicz.

**Writing – original draft:** Isabel Krug, Stephanie Fung, Shanshan Liu, Olivia McConchie.

**Writing – review & editing:** Isabel Krug, Shanshan Liu, Janet Treasure, Chia Huang, Kim Felmingham, Matthew Fuller-Tyszkiewicz.

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
