## [Decision Letter · Decision Letter 0]

19 Feb 2024

PONE-D-23-32813The Impact of Oxytocin on Emotion Recognition and Trust: Does Disordered Eating Moderate these Relationships?PLOS ONE

Dear Dr. Krug,

Thank you for submitting your manuscript to PLOS ONE. After careful consideration, we feel that it has merit but does not fully meet PLOS ONE’s publication criteria as it currently stands. Therefore, we invite you to submit a revised version of the manuscript that addresses the points raised during the review process.

We look forward to receiving your revised manuscript.

Kind regards,

Hidenori Yamasue, M.D., Ph.D.

Academic Editor

PLOS ONE

Journal Requirements:

3. In the online submission form, you indicated that "Additional data is available upon request."

4. Ethics statement only appears at the end of the manuscript:

Your ethics statement should only appear in the Methods section of your manuscript. If your ethics statement is written in any section besides the Methods, please move it to the Methods section and delete it from any other section. Please ensure that your ethics statement is included in your manuscript, as the ethics statement entered into the online submission form will not be published alongside your manuscript.

Reviewers' comments:

Reviewer's Responses to Questions

**Comments to the Author**

1. Is the manuscript technically sound, and do the data support the conclusions?

Reviewer #1: Yes

Reviewer #2: Partly

2. Has the statistical analysis been performed appropriately and rigorously? 

Reviewer #1: Yes

Reviewer #2: No

3. Have the authors made all data underlying the findings in their manuscript fully available?

Reviewer #1: Yes

Reviewer #2: No

4. Is the manuscript presented in an intelligible fashion and written in standard English?

Reviewer #1: Yes

Reviewer #2: Yes

5. Review Comments to the Author

Reviewer #1: Comments to the author

Manuscript Number: PONE-D-23-32813

Title: The Impact of Oxytocin on Emotion Recognition and Trust: Does Disordered Eating Moderate these Relationships?

Authors: Isabel Krug, Stephanie Fung, Shanshan Liu, Janet Treasure, Chia Huang, Kim Felmingham, Matthew Fuller-Tyszkiewicz & Olivia McConchie

Overview and comments:

This study examined the influence of oxytocin administration on various psychological factors, including emotion recognition, trusting behavior, affect, anxiety, and body satisfaction. Participants (149 female adults) were randomly assigned to receive either oxytocin or a placebo intranasal spray. The results indicate that individuals in the oxytocin group performed better in recognizing positive emotions but showed no significant effects in recognizing negative emotions, trusting behavior, state anxiety or state body image statisfaction. The severity of eating disorder symptoms did not moderate these effects. Overall, the study contributes to the existing literature on oxytocin and eating disorders, emphasizing its role in emotion recognition and positive affect. However I did note some minor comments below.

Minor comments

Page 4, line 81: it would be helpful to have more demographics from these studies (gender, age,..) to identify possible influencing factors.

Page 8, line 189: "ed", probably a typo in the sentence.

Page 8, line 190: I lack some background knowledge on oxytocin/ED and body satisfaction in the introduction.

Page 8, line 196: Now it seems like you expect them to have 'better performance at a trust game', but you would expect 'increased trust', not better game performance, right?

Page 8, 199-204: this is basically repeating the sentence before, seems redundant.

Page 9, line 228: Did oral contraceptives affect the results? What about non-oral contraceptives, menstrual cycle (or menopause)?

Page 11, line 271: Elaborate on why the 'intranasal spray' section was interleaved in the task and questionnaire sections.

Page 13, line 304: What was the scientific argumentation of not including baseline measures of these primary outcome measures?

Page 15, 358: So all participating investigators did have access to the unblinding?

Page 16, 399: Was this significantly different, seeing it is more than double?

Reading further, I noticed that the EAT scores indeed significantly differed between the intervention groups (p=0.02). This should investigated further and highlighted more in the manuscript.

Page 17, line 407: ‘Preliminary analyses’: are these not the final analyses?

Page 17, line 416: More informative subtitles would be appreciated.

Page 17, line 420: ‘Post-manipulation’: do the authors mean post-intervention?

Page 18, line 449: surprised and neutral, are these considered as positive emotions?

Page 19, line 468-470: I lack some references here.

Page 22, 525-526: “the current study theorised that if state anxiety decreased following an oxytocin intervention state body satisfaction would increase.” This was not found though, so this hypothesis might be more relevant to mention in the methods or even in the introduction.

Reviewer #2: The authors aimed to investigate the effect of oxytocin on emotion recognition, trust, body image, affect, and anxiety and whether eating disorder (ED) symptoms moderated any of these relationships. The current double-blind between subjects placebo controlled randomized trial of single dose oxytocin showed that the oxytocin-administered group exhibited better overall performance on the emotion recognition task and a decline in state positive affect than the placebo-administered group. However, these effects were not moderated by ED symptom severity.

The current study aim to test modulation of ED tendency on effect of oxytocin on social behavior / cognition seems to be interesting and relevant. However the following issues should be clarified or addressed:

The current Introduction and Discussion are too long. Especially, the Introduction should be shorten to clearly state the interesting, novelty and relevance of current study aim.

How to exclude candidates with psychiatric conditions. Please clarify the definition of psychiatric condition and the methodology to diagnose them.

The experiments for current study and data collection were conducted from Oct 2016 to Dec 2017. It is noted the long delay to submit the paper since the last data collection. When the data analyzed?

In page 11, it was described that the placebo was normal saline. It should be likely to differentiate the placebo and oxytocin for the participants because of their different smells. Please specify and report the degree to keep blindness among the participants.

Please consider to employ appropriate correction for multiple comparisons in the statistical analyses.

How to estimate the required sample size. Please clarify the methodology to decide the target sample size.

The results for the analyses of modulating effect of ED symptoms on efficacy of oxytocin should be reported in more details.

6. PLOS authors have the option to publish the peer review history of their article (what does this mean?). If published, this will include your full peer review and any attached files.

Reviewer #1: **Yes: **Matthijs Moerkerke

Reviewer #2: No

---

## [Author Response · Author response to Decision Letter 0]

18 Apr 2024

PONE-D-23-32813

The Impact of Oxytocin on Emotion Recognition and Trust: Does Disordered Eating Moderate these Relationships?

PLOS ONE 

Dear Prof. Hidenori Yamasue

Thank you for providing us with yours and the reviewers’ suggestions as to how we can improve our manuscript entitled: PONE-D-23-32813 " The Impact of Oxytocin on Emotion Recognition and Trust: Does Disordered Eating Moderate these Relationships?“

We have carefully reviewed the comments and have revised the manuscript accordingly. In the manuscript, we have highlighted with track changes all the changes we have undertaken. Our responses to the reviewers’ concerns are given in a point-by-point manner below.

Reviewer #1 

Minor comments

Comment 1: Page 4, line 81: it would be helpful to have more demographics from these studies (gender, age,..) to identify possible influencing factors.

Response 1: This information has now been deleted because one of reviewer 2 requested/recommended to shorten the introduction. 

Comment 2: Page 8, line 189: "ed", probably a typo in the sentence.

Response 2: Thanks for spotting this. We have fixed the typo in text.

Comment 3: Page 8, line 190: I lack some background knowledge on oxytocin/ED and body satisfaction in the introduction.

Response 3: We’ve added a few lines addressing this under ‘the Current study’. The new information added now reads as:” Specifically, given the co-occurrence of increased body dissatisfaction and state anxiety [7] in females with high levels of disordered eating and that oxytocin has been shown to decrease state anxiety [8], the current study theorised that if state anxiety decreased following an oxytocin intervention state body satisfaction would increase.”

Comment 4: Page 8, line 196: Now it seems like you expect them to have ‘better performance at a trust game’, but you would expect ‘increased trust’, not better game performance, right?

Response 4: Thank you for pointing this out. We have now rephrased this sentence. It now reads as: “We therefore hypothesised that compared to the placebo condition, (Hypothesis 1) participants in the oxytocin condition would score higher on emotion recognition and increased trust, and (Hypothesis 2) participants in the oxytocin condition would report decreased state anxiety and negative affect and increased state body satisfaction at post-intervention assessment.”

Comment 5: Page 8, 199-204: this is basically repeating the sentence before, seems redundant.

Response 5: Thank you – we edited this section for conciseness.

Comment 6: Page 9, line 228: Did oral contraceptives affect the results? What about non-oral contraceptives, menstrual cycle (or menopause)?

Response 6: Thank you for pointing out these two important influencing factors. Women who take oral contraceptives may sustain elevated oxytocin levels over prolonged periods. This heightened presence of oxytocin could diminish the impact of the hormone's typically moderate releases, potentially altering its usual physiological effects.

Garforth, B., Degnbol, H., Terris, E.T. et al. Elevated plasma oxytocin levels and higher satisfaction with life in young oral contraceptive users. Sci Rep 10, 8208 (2020). https://doi.org/10.1038/s41598-020-64528-w

Similarly, oxytocin plays a crucial role in both psychological and physiological processes, which often display distinct patterns based on sex, suggesting an interaction between oxytocin and sex hormones. Additionally, the levels of oxytocin may vary throughout the menstrual cycle, in tandem with shifts in the concentrations of female sex hormones.

Engel S, Klusmann H, Ditzen B, Knaevelsrud C, Schumacher S. Menstrual cycle-related fluctuations in oxytocin concentrations: A systematic review and meta-analysis. Front Neuroendocrinol. 2019 Jan; 52:144-155. doi: 10.1016/j.yfrne.2018.11.002. Epub 2018 Nov 17. PMID: 30458185.

Surprisingly only 22% of the current sample used the oral contraceptive pill and only 29% of the participants reported that they were in days 3 to 12 of their menstrual cycle.

We have provided more details about these variables that were reported in table 1 now in the text on Page 17. The new information now reads as:” Table 1 shows that the two groups were generally equivalent at baseline; chi-square tests confirmed no significant differences between the oxytocin and placebo groups, for the demographic (ethnicity, main language spoken at home, paid employment, education, marital status, and clinical variables (use of the contraceptive pill, other medication, self-reported psychiatric diagnoses, current and lifetime ED, day 3 to12 of menstrual cycle). However, participants in the oxytocin group scored significantly higher on eating pathology than the control group (32% vs 15%). Independent samples t-tests revealed no significant differences between groups for age and BMI. Fourteen percent of the sample (N = 21) had a high EAT-26 total score, and 10% of the sample (N = 15) reported having a history of an ED, with similar ratios across the two groups. There were only 3 participants with a lifetime self-reported psychiatric condition other than an ED (2 with mixed depression/anxiety and 1 with anxiety). Only 22% of the overall sample were currently using the contraceptive pill and 29% were currently on day 3 to 12 of their menstrual cycle.”

We have now outlined these influencing factors and the fact that we only assessed oral contraceptives as two additional limitations in the discussion. This section on Page 26 now reads as:” Finally, our sample were young females of whom 22% took oral contraceptives. It should be noted that women who take oral contraceptives may sustain elevated oxytocin levels over prolonged periods. This heightened presence of oxytocin could have diminished the impact of the moderate releases of oxytocin in our oxytocin group, potentially altering its usual physiological effects in this group [68]. Furthermore, we only asked about oral contraceptive usage and therefore do not know what percentage of our sample used other non-oral contraceptives. Similarly, it should be acknowledged that oxytocin may vary throughout the menstrual cycle, in tandem with shifts in the concentrations of female sex hormones and these fluctuations could have impacted our findings [69].

We don’t believe that menopause is an issue in the current sample since the mean age of our sample was 18.79.

Comment 7: Page 11, line 271: Elaborate on why the 'intranasal spray' section was interleaved in the task and questionnaire sections.

Response 7: The participants taking part in this experiment received research credits in exchange for taking part in the current study. This is why all the assessment had to be done in one session. Trait-based and sociodemographic questions were strategically placed at the beginning of the assessment, prior to the intervention, under the premise that these variables are independent and should not vary because of the intervention. We have outlined this on Page 15 (please refer to page numbers in the manuscript version with tracked changes):” Next, participants completed the immediately post-intervention measures, involving the demographic questionnaire and the trait measures EAT-26 (considered independent from the effects of the intervention).”

Comment 8: Page 13, line 304: What was the scientific argumentation of not including baseline measures of these primary outcome measures?

Response 8: The scientific argumentation of not including baseline measures of these primary outcome measures was that for these tasks practice effects can be found and also including a baseline assessment would have made it very difficult to disentangle the practice effects from the intervention effects. We have made this clearer on this page by adding the following sentence: “The TASIT-R and trust game were administered exclusively post-intervention, rather than at baseline, to prevent potential practice effects on these tasks from confounding the actual effects of the oxytocin or saline nasal sprays.”

Comment 9: Page 15, 358: So, all participating investigators did have access to the unblinding?

Response 9: No, only the principal investigator (first author) had access to the unblinding in case a participant a negative reaction to the intervention. The authors who were involved with data collection (O.M and S.F) did not have access to the unblinding. We have made this clearer on Page 15 by adding the following text:” The investigators (O.M and S.F) involved in data collection were blinded.” These student investigators were also not blinded when they conducted the analyses. This information was /isoutlined in the manuscript.

Comment 10: Page 16, 399: Was this significantly different, seeing it is more than double? Reading further, I noticed that the EAT scores indeed significantly differed between the intervention groups (p=0.02). This should be investigated further and highlighted more in the manuscript.

Response 10: We have clarified in text that this is significantly different for eating pathology. This information about significance levels is also contained in Table 1. While EAT scores might ordinarily be considered as a potential covariate or moderator to add for further analyses (to assess impact of this baseline group difference), we do not do so in the current study since EAT scores are already planned for use in analyses (Hypothesis 3). Given the lack of moderating effect of EAT scores, it appears that the group difference may not have had much effect in the current study. It is also reassuring that there were no other significant differences observed (see Table 1 and preliminary analysis subsection).

Comment 11: Page 17, line 407: ‘Preliminary analyses’: are these not the final analyses?

Response 11: We had used the terms ‘Preliminary Analyses’ and ‘Main Analyses’ to distinguish analyses for context (former) and those that directly address research hypotheses (latter). To make this clearer we have tried to make these titles more informative by adding the following titles:” 1.) Preliminary Analyses: Tests of differences pre-intervention and 2.) Main analyses: Hypotheses testing.

Comment 12: Page 17, line 416: More informative subtitles would be appreciated. 

Response 12: We’ve edited the subtitle to highlight the key results.

This new subtitle is now worded as:” Hypothesis 1 - Oxytocin and Better Performance on the Recognition of Positive Emotions” We have also rephrased the subtitle for Hypothesis 3 testing. It now reads as:” Hypothesis 3 – Moderation Effects of Eating Pathology”.

Comment 13: Page 17, line 420: ‘Post-manipulation’: do the authors mean post-intervention?

Response 13: We’ve changed the wording to post-intervention for consistency.

Comment 14: Page 18, line 449: surprised and neutral, are these considered as positive emotions?

Response 14: Yes, according to the TASIT-R, which is a well validated measure they are. 

Comment 15: Page 19, line 468-470: I lack some references here.

Response 15: We’ve added relevant references to support the statement. “Our finding is also consistent with research involving healthy individuals [28], which reported that a single dose of oxytocin administration improved emotion recognition performance in both male and female participants [9, 29].”

Comment 16: Page 22, 525-526: “the current study theorised that if state anxiety decreased following an oxytocin intervention state body satisfaction would increase.” This was not found though, so this hypothesis might be more relevant to mention in the methods or even in the introduction.

Response 16: We’ve moved statements relating to this to the introduction section. 

Reviewer #2: 

Comment 18: The current Introduction and Discussion are too long. Especially, the Introduction should be shortened to clearly state the interesting, novelty and relevance of current study aim.

Response 18: We’ve made edits to shorten these sections and to highlight the focus of the study.

Comment 19: How to exclude candidates with psychiatric conditions. Please clarify the definition of psychiatric condition and the methodology to diagnose them.

Response 19: This information was already included in the participant section on Page 9:” Exclusion criteria (self-reported) included: a.) males; b.) aged <18 or >30; c.) being pregnant/trying to become pregnant; d.) breast feeding/breast-fed in the last two months; e.) heavy smokers (> 5 cigarettes per day) and/or heavy drinkers (>14 standard drinks per week); f.) having a current substance use disorder or other psychiatric condition other than an ED and g.) a history of heart disease, cardiac disorder, neurological disorder, or allergic reaction to preservatives (food additives).” We have slightly rephrased this section so that it is now hopefully clear that the other psychiatric condition refers to any other psychiatric condition than an ED.

None of the participants presented with a current psychiatric diagnosis and only 3 with a lifetime diagnosis (two with depression and anxiety and one with only anxiety). 

We have added this information to the participant section on Page 17-18:” There were only 3 participants with a lifetime self-reported psychiatric condition other than an ED (2 with mixed depression/anxiety and 1 with anxiety).”

Comment 20: The experiments for current study and data collection were conducted from Oct 2016 to Dec 2017. It is noted the long delay to submit the paper since the last data collection. When was the data analysed?

Response 20: The lag in publishing this data was because the primary author had been on parental leave 3 times since the data had been collected; once in 2017, once in 2019 and once in 2022. Taking care of three small children under the age of 6 in addition to working full time in a teaching and research position and having clinical responsibilities, unfortunately did not allow for a faster turn around and write of this manuscript. The data was analysed in 2017/2018 by two Master students, and they used these analyses to write up their Master thesis. The data was afterwards re-analysed by a seasoned statistician for the purpose of this publication. 

Comment 21: In page 11, it was described that the placebo was normal saline. It should be likely to differentiate the placebo and oxytocin for the participants because of their different smells. Please specify and report the degree to keep blindness among the participants.

Response 21: Neither the Oxytocin nor the Saline intervention had a smell and therefore it is extremely unlikely that the non-existing smell could have influenced the findings. We have added this information to this section by including the following information:” The intranasal sprays, which were both odorless, were contained in identical bottles with labels stating the participant code (i.e., 1, 2, 3 etc.). This number was linked to the condition (i.e., oxytocin or placebo intervention).”

Comment 22: Please consider employing appropriate correction for multiple comparisons in the statistical analyses.

Response 22: We have elaborated on our approach to multiple comparisons in the Data Analysis plan section. The following information was added on Page 17: “Although a range of outcome measures was tested, we retained alpha criterion at .05 rather than correcting for multiple comparisons. Given the paucity of studies exploring the effects of oxytocin in this context, there is a risk that correction for Type I error rate increases risk of Type II errors [56-58), prematurely terminating a viable line of inquiry.” 

[56] Feise, R.J. (2002). Do multiple outcome measures require p-value adjustment? BMC Medical Research Methodology, 2, 8. Doi: 10.1186/1471-2288-2-8

[57] Rothman, K.J. (1990). No adjustments are needed for multiple comparisons. Epidemiology, 1(1), 43-46.

[58] Rothman, K.J. (2010). Curbing type 1 and type 2 errors. European Journal of Epidemiology, 25(4), 223-224. Doi: 10.1007/s10654-010-9437-5

We have also added this as another potential limitation in the discussion on Page 26:” Fourth, while the observed effect sizes and statistically significant results for the emotion recognition and state positive emotion findings are promising, it is important to approach these findings with caution. The possibility of Type I errors, arising from multiple comparisons,

---

## [Decision Letter · Decision Letter 1]

22 Apr 2024

PONE-D-23-32813R1The Impact of Oxytocin on Emotion Recognition and Trust: Does Disordered Eating Moderate these Relationships?PLOS ONE

Dear Dr. Krug,

Thank you for submitting your manuscript to PLOS ONE. After careful consideration, we feel that it has merit but does not fully meet PLOS ONE’s publication criteria as it currently stands. Therefore, we invite you to submit a revised version of the manuscript that addresses the points raised during the review process.

We look forward to receiving your revised manuscript.

Kind regards,

Hidenori Yamasue, M.D., Ph.D.

Academic Editor

PLOS ONE

Journal Requirements:

Reviewers' comments:

Reviewer's Responses to Questions

**Comments to the Author**

1. If the authors have adequately addressed your comments raised in a previous round of review and you feel that this manuscript is now acceptable for publication, you may indicate that here to bypass the “Comments to the Author” section, enter your conflict of interest statement in the “Confidential to Editor” section, and submit your "Accept" recommendation.

Reviewer #1: (No Response)

Reviewer #2: (No Response)

2. Is the manuscript technically sound, and do the data support the conclusions?

Reviewer #1: Yes

Reviewer #2: Partly

3. Has the statistical analysis been performed appropriately and rigorously? 

Reviewer #1: Yes

Reviewer #2: No

4. Have the authors made all data underlying the findings in their manuscript fully available?

Reviewer #1: Yes

Reviewer #2: Yes

5. Is the manuscript presented in an intelligible fashion and written in standard English?

Reviewer #1: Yes

Reviewer #2: Yes

6. Review Comments to the Author

Reviewer #1: Comment 11: Page 17, line 407: ‘Preliminary analyses’: are these not the final analyses?

Response 11: We had used the terms ‘Preliminary Analyses’ and ‘Main Analyses’ to

distinguish analyses for context (former) and those that directly address research hypotheses

(latter). To make this clearer we have tried to make these titles more informative by adding the

following titles:” 1.) Preliminary Analyses: Tests of differences pre-intervention and 2.) Main

analyses: Hypotheses testing.

I believe 'preliminary' might not be the most suitable term to use.

Comment 14: Page 18, line 449: surprised and neutral, are these considered as positive

emotions?

Response 14: Yes, according to the TASIT-R, which is a well validated measure

they are.

I'm still concerned about the classification of "surprised" and "neutral" emotions as positive in the TASIT-R, as this seems to oversimplify the complex nature of these emotions (not-negative does not mean positive). Emotions like surprise can vary greatly depending on context—being pleasant when associated with good news, but negative if linked to shock or fear. Similarly, neutrality might be seen positively as calmness or negatively as indifference. The TASIT-R, while well-validated for assessing social perception, might not fully capture these nuances, especially given that emotional reactions can be highly subjective and context-dependent. Some addional nuance would be appropriate here.

Reviewer #2: Most of my comments were addressed by the revision. However, some additional revisions should further be considered in the following points:

Response 19: My previous comment on the initial submission mean that the authors should clarify how to confirm that none of the participants presented with a current psychiatric diagnosis. Did you utilize diagnostic tools such as SCID? Who conducted the assessment of psychiatric diagnosis? Psychiatrist? Psychologist? Or just self report? Please clarify and report them.

Response 20: It should be helpful to report the time period for analyses.

Response 21: It was known that Syntocinon have some smells. Therefore, most of previous studies have employed placebo containing all inactive ingredient to avoid any subjective effects of the substances other than those caused by oxytocin. Utilizing saline as placebo should be considered as a limitation.

7. PLOS authors have the option to publish the peer review history of their article (what does this mean?). If published, this will include your full peer review and any attached files.

Reviewer #1: No

Reviewer #2: No

---

## [Author Response · Author response to Decision Letter 1]

30 Apr 2024

PONE-D-23-32813 R1

The Impact of Oxytocin on Emotion Recognition and Trust: Does Disordered Eating Moderate these Relationships?

PLOS ONE 

Dear Prof. Hidenori Yamasue

Thank you for providing us with yours and the reviewers’ suggestions as to how we can improve our manuscript entitled: PONE-D-23-32813R1 " The Impact of Oxytocin on Emotion Recognition and Trust: Does Disordered Eating Moderate these Relationships?“

We have carefully reviewed the comments and have revised the manuscript accordingly. In the manuscript, we have highlighted with track changes all the changes we have undertaken. Our responses to the reviewers’ concerns are given in a point-by-point manner below.

Reviewer #1: 

Comment 1: I believe 'preliminary' might not be the most suitable term to use.

Response 1: We have changed the wording for these analyses. The subheading now specifies these analyses as:” Differences in outcome variables between the oxytocin and control group at pre-intervention.” We have also deleted the “Main Analyses” and now only refer to “Hypotheses Testing”.

Comment 2: Page 18, line 449: surprised and neutral, are these considered as positive

emotions?

I'm still concerned about the classification of "surprised" and "neutral" emotions as positive in the TASIT-R, as this seems to oversimplify the complex nature of these emotions (not-negative does not mean positive). Emotions like surprise can vary greatly depending on context—being pleasant when associated with good news, but negative if linked to shock or fear. Similarly, neutrality might be seen positively as calmness or negatively as indifference. The TASIT-R, while well-validated for assessing social perception, might not fully capture these nuances, especially given that emotional reactions can be highly subjective and context-dependent. Some additional nuance would be appropriate here.

Response 2: We appreciate the feedback from the reviewer and agree that these emotions can have a range of different connotations. Unfortunately, we can’t change the scoring of the TASIT-R. We have however, outlined this as a further limitation in the discussion section. This new limitation on Page 24 now reads as:” Fourth, it is important to acknowledge that the perception of emotions, such as surprise, can exhibit considerable variability contingent upon the circumstances. For instance, surprise may evoke positive sentiments when linked with delightful news, whereas it may evoke negative reactions when associated with shock or fear. Similarly, neutrality can be construed positively, reflecting a sense of calmness, or negatively, suggesting indifference. Despite the comprehensive validation of the TASIT-R for assessing social perception, it may not fully capture these intricate nuances, primarily due to the subjective and context-dependent nature of emotional responses.”

Reviewer #2: 

Comment 3: My previous comment on the initial submission mean that the authors should clarify how to confirm that none of the participants presented with a current psychiatric diagnosis. Did you utilize diagnostic tools such as SCID? Who conducted the assessment of psychiatric diagnosis? Psychiatrist? Psychologist? Or just self-report? Please clarify and report them.

Response 3: These diagnoses were based on self-report. We had already included this previously on Page 7 (self-reported) has now been highlighted to clarify this: ” Exclusion criteria (self-reported) included: a.) males; b.) aged <18 or >30; c.) being pregnant/trying to become pregnant; d.) breast feeding/breast-fed in the last two months; e.) heavy smokers (> 5 cigarettes per day) and/or heavy drinkers (>14 standard drinks per week); f.) having a current substance use disorder or any other psychiatric condition other than an ED and g.) a history of heart disease, cardiac disorder, neurological disorder, or allergic reaction to preservatives (food additives). Exclusionary criteria were selected based on safety reasons (i.e., oxytocin can induce contractions in pregnancy) and on findings that these factors can interfere with the effectiveness of oxytocin [48].”

Comment 4: It should be helpful to report the time period for analyses.

Response 4: The time periods for the analyses have now been added to page 14. “Initial analyses were first conducted in 2017/2018 for two independent Master theses (O.M and S.F.) and later re-run for the current article in 2022 by a seasoned statistician (M.F).”

Comment 5: It was known that Syntocinon have some smells. Therefore, most of previous studies have employed placebo containing all inactive ingredient to avoid any subjective effects of the substances other than those caused by oxytocin. Utilizing saline as placebo should be considered as a limitation.

Response 5: Thank you for this further suggestion. We have now added this as a further limitation in the discussion. On Page 24/25 we have added the following information: “Seventh, the choice of saline as a placebo in this study may introduce a further limitation, as it may not completely replicate the inert properties of a placebo containing all inactive ingredients. Future studies should therefore employ placebos containing all inactive ingredients to mitigate potential subjective effects of substances other than oxytocin.”

We are grateful to the Editor and reviewers for their time and suggestions which have greatly improved the manuscript. We hope the revised version is now suitable for publication in PloSONE and look forward to hearing from you in due course. 

Yours sincerely on behalf of the authors

---

## [Editor Report · Decision Letter 2]

2 May 2024

The Impact of Oxytocin on Emotion Recognition and Trust: Does Disordered Eating Moderate these Relationships?

PONE-D-23-32813R2

Dear Dr. Krug,

We’re pleased to inform you that your manuscript has been judged scientifically suitable for publication and will be formally accepted for publication once it meets all outstanding technical requirements.

Kind regards,

Hidenori Yamasue, M.D., Ph.D.

Academic Editor

PLOS ONE

---

## [Editor Report · Acceptance letter]

21 May 2024

PONE-D-23-32813R2 

PLOS ONE

Dear Dr. Krug, 

I'm pleased to inform you that your manuscript has been deemed suitable for publication in PLOS ONE. Congratulations! Your manuscript is now being handed over to our production team.

Kind regards, 

on behalf of

Dr. Hidenori Yamasue 

Academic Editor

PLOS ONE